# Conducting Polymers, Hydrogels and Their Composites: Preparation, Properties and Bioapplications

**DOI:** 10.3390/polym11020350

**Published:** 2019-02-17

**Authors:** Monika Tomczykowa, Marta Eliza Plonska-Brzezinska

**Affiliations:** Department of Organic Chemistry, Faculty of Pharmacy with the Division of Laboratory Medicine, Medical University of Bialystok, Mickiewicza 2A, 15-222 Bialystok, Poland; monika.tomczyk@umb.edu.pl

**Keywords:** conducting polymer, composite, bioapplication

## Abstract

This review is focused on current state-of-the-art research on electroactive-based materials and their synthesis, as well as their physicochemical and biological properties. Special attention is paid to pristine intrinsically conducting polymers (ICPs) and their composites with other organic and inorganic components, well-defined micro- and nanostructures, and enhanced surface areas compared with those of conventionally prepared ICPs. Hydrogels, due to their defined porous structures and being filled with aqueous solution, offer the ability to increase the amount of immobilized chemical, biological or biochemical molecules. When other components are incorporated into ICPs, the materials form composites; in this particular case, they form conductive composites. The design and synthesis of conductive composites result in the inheritance of the advantages of each component and offer new features because of the synergistic effects between the components. The resulting structures of ICPs, conducting polymer hydrogels and their composites, as well as the unusual physicochemical properties, biocompatibility and multi-functionality of these materials, facilitate their bioapplications. The synergistic effects between constituents have made these materials particularly attractive as sensing elements for biological agents, and they also enable the immobilization of bioreceptors such as enzymes, antigen-antibodies, and nucleic acids onto their surfaces for the detection of an array of biological agents. Currently, these materials have unlimited applicability in biomedicine. In this review, we have limited discussion to three areas in which it seems that the use of ICPs and materials, including their different forms, are particularly interesting, namely, biosensors, delivery of drugs and tissue engineering.

## 1. Introduction

Electronically conducting polymers (intrinsically conducting polymers, ICPs) are a class of organic polymers possessing high electronic conductivity that were first synthesized as early as 1862 [1]. Letheby prepared polyaniline (PANI) via the anodic oxidation of aniline, and this polymer was conductive and showed electrochromic behaviour [2]. This field has not developed extensively since the mid-1970s, when a new class of polymers capable of acquiring charge was discovered. The preparation of polyacetylene (PA) and the discovery of its conductivity after “doping” launched this new field of research. Heeger, MacDiarmid and Shirakawa received the Nobel Prize in Chemistry in 2000 “for the discovery and development of electronically conductive polymers” [3,4,5].

Conducting polymers (CPs) are similar to metals and semiconductors due to their electrical and optical properties, while retaining the properties of common polymers, such as easy and inexpensive synthesis and flexibility [1]. These materials are versatile because their properties can be easily modulated by surface functionalization and/or doping [6]. The fundamental nature of charge propagation in CPs is based mainly on the following two mechanisms: (i)the movement of delocalized electrons through conjugated systems in ICPs (e.g., polypyrrole (PPy) and PANI), and(ii)the transport of electrons via an electron exchange reaction (electron hopping) between neighbouring redox sites in redox polymers [1].

The mode of charge propagation is linked to the chemical structure of the polymer. Due to this mode, CPs can be classified into electron-conducting polymers (redox polymers and ICPs) and proton (ion)-conducting polymers [1].

The conductive properties of CPs make them an important class of materials for a wide range of applications [7,8], mainly in energy storage [9,10,11,12] electronic and photovoltaic devices [13], electrochromic displays [14,15], electrocatalysis and photocatalysis [16,17], and sensors [18,19,20,21,22,23], etc. CPs have garnered increasing attention in biomedicine because they can convert different types of signals into electrical signals. Since the 1980s, when it was found that these materials are compatible with many biological molecules, their biomedical applications have expanded greatly [24]. Due to their excellent biocompatibility, these “smart materials” may be used in different areas of biomedicine [25,26], such as cell (cell growth and cell migration) and tissue engineering, biosensors, drug and gene delivery systems, artificial muscles, and diagnostic applications [27,28,29,30,31,32,33], etc.

## 2. Synthesis, Physicochemical and Biological Properties of Conducting Polymers, Conductive Hydrogels and Their Composites

### 2.1. Undoped Conducting Polymers

#### 2.1.1. Redox-Polymers

Organic CPs, which contain electrostatically and spatially localized redox sites and in which electrons are transported by an electron exchange reaction between redox neighbouring sites, are called redox polymers. These polymers may be divided into the following: (i)polymers that contain covalently attached redox sites (organic or organometallic molecules), and,(ii)ion-exchange polymeric systems (polyelectrolytes) in which the redox active ions are held by electrostatic binding [1].

The first group, in which the redox group is incorporated into the chain, is exemplified by poly(viologens), while the second group with pendant redox groups is exemplified by poly(tetrathiafulvalene), quinone polymers (Scheme 1, structure **1**) and poly(vinylferrocene). Typical examples of ion-exchange polymers containing electrostatically bound redox centres are perfluorinated sulfonic acids (Nafion) (Scheme 1, structure **2**), poly(styrene sulfonate) and poly(4-vinylpyridine). Redox polymers are not frequently used in the biomedical areas that will be discussed in this paper; therefore, they will not be mentioned in more detail in this review.

#### 2.1.2. Electronically Conducting Polymers (Intrinsically Conducting Polymers)

In the case of ICPs, delocalized electrons occur through conjugated systems, i.e., conjugated π−polymers. ICPs are electrically conductive, meaning delocalized π−electrons move freely within the backbone to construct an electrical pathway for mobile charge carriers [34]. The electron hopping is based on interchain conduction and defects, which usually lead to reorganization of the bonds of the polymers. The structures of these polymers mainly contain benzenoid or nonbenzenoid (mostly amines) and heterocyclic compounds. The polymers are electroactive when they are partially oxidized or, less frequently, reduced. Typical representatives of ICPs are PANI (Scheme 2), PPy (Scheme 1, structure **3**), poly(3,4-ethylenedioxythiophene) (PEDOT) (Scheme 1, structure **4**), polythiophene (PT), polyphenazine (PPh), PA, polycarbazole (PCz), poly(*p*-phenylene) (PPP) and their derivatives. Some examples of the most widely used ICPs are presented in Table 1 [7].

Polymers can be prepared using chemical and/or electrochemical methods of polymerization [62,63,64]. A chemical route is recommended if large amounts of polymer are needed. For relevant applications, electrochemical methods are preferable because good-quality material is formed [1]. The reaction is usually an oxidative polymerization, although reductive polymerization is also possible [63,65]. Using electrochemical methods, aromatic [66], benzoid (e.g., aniline) [67,68] or nonbenzoid (e.g., 1-aminoanthracene) [69], and heterocyclic compounds (e.g., pyrrole [70,71], thiophene [72]) can be polymerized. Upon polymerization, the oxidized polymer backbone generally carries a positive charge, which is balanced by a negatively charged counter-ion [18]. These negatively charged molecules, called dopants, become part of the formed ICP. Dopants may be an anion, either small (e.g., Cl^−^, ClO_4_^−^) or large (e.g., polystyrene sulfonate (PSS)) [18,73,74,75].

PA is a linear polyene chain [–(HC=CH)_n_–] that may be synthesized by a Ziegler–Natta catalyst or by radiation methods [3,4,5,62]. The pristine form of PA is easily oxidized in air and is also sensitive to humidity. Each repeated unit of hydrogen can be replaced by one or two substitutes, giving monosubstituted or disubstituted PAs, respectively. PA is also called acetylene black (AB) due to its preparation method [76,77]. AB is produced by the controlled combustion of acetylene under air. AB is usually used for the preparation of carbon paste electrodes [78,79], for construction of biosensors [80], and for inclusion of enzymes [81]. Electrodes containing AB show excellent biocompatibility, high electrical conductivity and a large specific surface area, which results in electrochemical sensors with good sensitivity [80,82,83,84]. Pristine AB or its doped form have been used for the detection of glucose oxidase [81], colchicine [85], monoamine neurotransmitters and their metabolites [86].

Polythiophene (PT) can be synthesized by electrochemical and chemical methods [72,87,88,89]. PT has a high stable conductivity (10^3^ S cm^−1^) that varies with the type of dopant and polymerization [90,91]. One of the most important properties of this polymer is its transparency, which is a function of its dilution and in turn affects its conductivity [92,93]. Dilution can be achieved by blending with a transparent polymer, grafting, copolymerization, plasma polymerization or electrochemical methods [92,94,95]. The conductivity of PT may also be improved by hybridization of the polymer chain with different dopants, for example, PSS, lithium perchlorate, tetraethylene glycol, and alkyl side chains [18,89,96,97], etc. One of the most popular PT derivatives is PEDOT (Scheme 1, structure **4**), characterized by very high conductivity, high electrochemical stability and a very narrow band gap [98,99,100,101]. Additionally, PEDOT is easily oxidized [101]. These features make PEDOT and its derivatives preferable in many electroanalytical applications [98], such as the detection of organic species (e.g., ascorbic acid, dopamine, uric acid) [102,103,104], metal ions [105,106], and inorganic species [107,108], etc.

PPy is one of the most frequently used ICPs for biomedical applications due to its high conductivity, biocompatibility, easy synthesis and environmental stability [25,109,110]. Similar to other ICPs, the conductivity of PPy is strongly connected with the structures of its polymer chains. The biocompatibility of PPy with nerve tissue was evaluated in vitro and in vivo [25,110,111]. An extraction solution of PPy powder, which was synthesized chemically, was tested for acute and subacute toxicity, pyretogen, haemolysis, allergen, quantitative measurement of cell viability, and micronuclei [25]. The results of this study indicated that PPy has good biocompatibility with rat peripheral nerve tissue. PPy may be synthesized by several chemical and electrochemical methods that preserve its high conductivity. Similar to other ICPs, its conductivity may be improved by its hybridization with other materials (myocytes, biotin, alginate, silk fibroin, tosylate ions, etc.) [110,112,113]. The unusual physicochemical properties of PPy and its biocompatibility may be applied in nerve regeneration, biomedicine or biosensors (Table 1) [25,109,110,114,115].

The undoped form of poly(*p*-phenylene) (PPP) is an insoluble powder that shows interesting thermal, optical electrical and chemical properties. Its ICPs, which have low molecular weights and with modified chains, are soluble in many solvents [116,117,118,119]. They may be synthesized by electrochemical [120] and chemical methods [116,121], for example, polycondensation with the Suzuki coupling method or catalytic polymerization. PPP frequently shows linearly dichromic fluorescence [122]. PPP and its derivatives are conventionally used in the form of thin films operating as active layers in light-emitting diodes, photodetectors, and other optoelectronic devices [121]. PPP may also be applied in dental applications and cell alignment (Table 1).

One of the most promising ICPs, next to PPy, is PANI. This polymer shows high electrical conductivity and great environmental stability [123,124,125]. Its synthesis is very easy and low-cost, and its conductivity depends on its oxidation state [64]. PANI chains consist of −p-coupled aniline units [61]. The combination of benzenoid and quinoid rings leads to different oxidation states for the PANI polymer: leucoemeraldine, emeraldine, and pernigraniline (Scheme 2) [126,127]. The green emeraldine form of PANI exhibits electroconductive properties (Table 1). Charge carrier mobility in the highly conductive emeraldine ranges from 10^−3^ to 10^−1^ cm^2^ V^−1^s^−1^ [128]. PANI conductivity may be optimized by increasing crystallinity and conjugation length of the polymer [129]. The conductivity of PANI is also strongly connected with the structures of the polymer chains. PANI chains can form one-dimensional (1D) structures (nanofibres, nanorods and nanotubes) [130,131,132], planar two-dimensional (2D) objects (e.g., ribbons, nanobelts and nanoplates) [61,133] and three-dimensional (3D) particles (microspheres, nanospheres and granules) [134,135,136,137]. Nanostructured PANI offers the possibility of enhanced performance for the fast transfer of electrons and can be envisioned for potential applications including supercapacitors [138,139], biosensors [61,140,141] and microelectronic devices [142].

PANI can be synthesized by several chemical (e.g., oxidative polymerization, via conventional free-radical polymerization, enzymatic synthesis) [129,137,143,144] and electrochemical methods [144,145]. PANI possesses controlled conductivity combined with ionic and proton conductivity, redox activity, electro- and solvatochromism, non-linear optical properties and paramagnetism [146]. Additionally, PANI is nontoxic, stable in aggressive chemical environments and has high thermal stability. These properties facilitate its application in biosensing, medicine and tissue engineering (Table 1).

Notably, the conductivity of all ICPs is strongly connected with the structures of their polymer chains. ICPs can be formed into aqueous gels, membranes, nanofibres, and nanowires, etc., mainly by physically compositing or forming co-networks. In recent years, some of the most interesting reports concern conductive hydrogels (conducting polymer hydrogels, CPHs) based on single component ICPs (PANI, PPy, Pt) as the continuous phase. Despite their unusual structures (some representative hydrogels/membranes of ICPs are presented in Figure 1) [141,147,148,149], the resulting CPHs show some novel properties, e.g., shape memory elasticity, fast functionalization with various guest objects, and fast removal of organic pollutants from aqueous solutions [148]. Furthermore, lightweight, elastic, and conductive organic sponges/hydrogels have been successfully applied in biomedical applications, which will be discussed in the following paragraphs.

### 2.2. Conducting Polymer Hydrogels

The term hydrogel usually refers to a network matrix made up of highly swollen and entangled polymers that contain significant amounts of water in their network (c.a. 90–95%). Hydrogels have a spatially cross-linked chain network composed of natural and/or synthetic hydrophilic polymer chains that absorb a large amount of water while maintaining a 3D structure [150,151], which creates great potential for their use for biomedical purposes [114,152,153]. CPH is a material that contains a CP (ICP or redox-polymer), frequently together with a supporting polymer. Due to their specific construction, CPHs exhibit some interesting properties, such as: high water content, softness, plasticity and mechanical integrity, porosity and high surface area specificity [154,155]. In addition, they are characterized by mixed electronic and ionic conductivity, redox activity, and conversion among conducting insulating forms in CPs. The most common conductive components of CPHs are PANI, PPy and PEDOT [144,154,156,157,158,159].

As already mentioned above, one of the most important properties of CPHs is their electrical conductivity resulting from the ease with which electrons leap between and inside the polymer chains [160]. In fact, conductivity is affected by several factors. These factors include (i) CP type, (ii) heterogeneity of the CPH structure, and (iii) different shapes and sizes of the CPH particles, as well as (iv) the presence of water bound by CPHs and (v) additional components, which affect conductivity [160]. Analogous to ICPs prepared by standard methods, the occurrence of p-orbitals in the conjugated system of double bonds facilitates the delocalization of electrons and their free movement between atoms. The second type of charge propagation in CPs, which affects conductivity, is electron hopping in redox polymers. Redox polymers are synthesized in a conducting (oxidized) form, which is stabilized by the addition of a dopant, neutralizing the CP charge. In most cases, it is a negative charge. A dopant additive causes destabilization of the CP skeleton at the moment of application of electric potential to CP by entering or leaving the polymer (depending on the polarity) [160,161]. In such a destabilized CP skeleton, the charge is able to flow through the polymer [162,163]. The electrical conductivity of most known and described CPHs reaches values up to 10 mS cm^−1^. As mentioned in a previous section, the mechanism of electrical conductivity in CPHs is similar to that observed in bulk or nanostructured ICPs. However, in the case of CPHs, their conductivity also strongly depends on the electrolyte solution, which is a factor of hydrogel hydration [164] and is considered to be one of the main factors responsible for the electrical conductivity of hydrogels.

The essential physical parameters of CPHs include (i) tacticity, (ii) glass transition temperature, (iii) swelling and (iv) water amount bounded in the system. These parameters are particularly important in relation to the properties of CPHs that are potential carriers of drugs or in TE.

The factor indicating the modalities of distribution of monomer units along the polymer chain is tacticity. Substitution of monomers in the polymer is possible in the following three different ways [165,166]:isotactic—all of the same type of substituents are ordered on the same side of the polymer chain;syndiotactic—the substituents are arranged alternately on both sides of the polymer chain;heterotactic—substituents are distributed along the polymer chain irregularly.

The glass transition temperature is another parameter describing hydrogels. This is the temperature at which, due to the increased viscosity of the polymer solution, it becomes a solid. The more regular the hydrogel structure, the lower the glass transition temperature value [165]. The degree of cross-linking and the method of water binding determine the level of swelling of the CPH. Water molecules are bound primarily by the most polar hydrophilic groups of the polymer. Larger spaces and pores of CPHs are often additionally filled with “free water” [165,167].

The degree of CPH cross-linking is defined as the ratio of the cross-linking agent moles to the moles of the repeating polymer units. A higher degree of CP cross-linking means a more compact structure, which results in the reduced mobility of polymer chains and ultimately reflects a lower degree of swelling [167]. The nature of the covalent bonds found in CPHs affects their mechanical properties. Reduced mechanical properties result in gels based on van der Waals forces or hydrogen bonding. CP and dopant amounts also change the mechanical properties of CPHs. The size and shape of clusters containing CP and their heterogeneous distribution in CPHs have an impact on the mechanical properties of CPHs.

The synthesis of CPHs can be carried out using natural or synthetic polymers [159,168,169]. Considering the advantages and disadvantages of both groups of polymers, synthetic polymers are preferred. This is because CP preparation must meet strictly defined needs/parameters.

CPHs may be obtained by different types of methods [141,170,171,172,173]. One method relies on mixing the hydrogel component with the conducting component. Another is based on the polymerization of CP monomers in a hydrogel matrix. One of the most commonly used methods is the preparation of a hydrogel containing a supporting polymer, which then forms a matrix for the synthesis of a CP [174]. This concept is presented on Figure 2A. CPH consist of CP built in a matrix created from water-soluble polymers, which are swollen with water or electrolyte solution (Figure 2A). This synthesis begins with the creation of a hydrogel matrix, in which the CP solution is then distracted. The matrix containing the CP monomers prepared in this way is suspended in an oxidant solution, which diffuses into the hydrogel. The CP is created after encountering the monomer. This kind of method is called interfacial polymerization (Figure 2A), and it is used to receive CPHs in which the CP role is fulfilled with PANI [175,176,177,178], PPy [179,180,181] and rarely with PEDOT [182,183,184,185,186]. A modified interfacial polymerization method has been used to obtain PANI [187] and PPy [188]. CPHs are also obtained as a result of the addition of CP powders into the reaction mixture, which is the basis for obtaining a hydrogel. Using this method, derivatives of PANI [189], as well as PPy [190] and PEDOT [191], may be synthesized. When the liquid is replaced with a gas, leaving the pores intact without destroying the porous structure, hydrogels are transformed into aero-sponge-like materials (Figure 2B). The scanning electron microscopic images presented in Figure 2B show a highly developed surface area of porous materials (hydrogels and aero-sponge-like materials), one from the crucial properties of hydrogels for their biomedical applications.

Due to the fact that ICPs are mostly hydrophobic, researches are continued on the use of various strategies to increase the applicability of CPs in the hydrogel matrix [8,192,193,194,195,196]. Well-defined architecture of nanostructures, including tunable filler configuration, size, density, specific surface area, increase in the number of hydrophilic groups are factors taken into account in designing and fabricating of CPHs [8]. Increasing the effective surface area of CPHs and reducing the interfacial tension between the aqueous phase and the CPHs, can be obtained by using different types of CPs (e.g., PPY or PT) with unique morphology and functionalities with different degree of dispersion [197,198]. In order to control the roughness geometry and wettability of ICPs incorporated into CPHs, chemical oxidative polymerization in the presence of porous membranes and nanofillers (nanoparticles, nanowires, nanofibers, etc.) are performed [193,194,195,199]. When the nature of the ICPs is not sufficient to induce the formation of nanostructures, surfactants can be employed to induce these nanostructures [200]. Controlling the surface morphology of CPHs can be relatively easily achieved by the proper selection of CP strategy such as chemical or electrochemical polymerization in solution or vapor phase [192,196].

The importance of individual functions is strictly connected with specific applications. CPHs are nontoxic and compatible with living tissue or cells [49,115,173,201]. CPH properties offer their use in various interesting fields, particularly in biomedicine and energy storage, and are among the most promising trends in materials science, which have been and are still widely studied [148,156,157,158,159,202]. Some common features of CPHs are discussed below, among which the sub-standard is their inhomogeneity. Therefore, the properties of this material group are closely related to hydrogel shape and size. This paradigm creates many difficulties in defining and comparing CPHs, which is often possible only after the final examination of a given CPH.

### 2.3. Conductive Polymer Nanocomposites

An effective way to improve the mechanical stability of ICPs is to create composites with nanoparticles or blends with other polymers that have better mechanical properties for their intended applications than their pristine analogues. Conductive polymer nanocomposites (ICP_comp_) combine the flexibility and/or conductivity of the polymer with the distinct properties of nanofillers. Figure 3 presents an overview of conductivity of CPs and their composites [203].

CPs may form well-defined porous nanostructures (for example, gels), where they translate the properties of their bulk forms and exhibit unusual chemical and physical properties owing to the dimensions of nanomaterials, for example, zero-dimensional (0D) nanoparticles [204,205], 1D fibres or 2D sheets. In this context, CPs may be an ideal matrix to introduce a new component due to their porous structures. The design and synthesis of ICP_comp_ inherits the advantages of each component and offers new features because of the synergistic effects of the composite’s components [206,207].

The ICP_comp_, taking into account further bioapplications, may be synthesized in the following two ways: (i)a two-step synthesis in which the gel network is created and then acts as a matrix for further modification, or,(ii)(a one-step method in which monomers are polymerized and cross-linked with other precursors in solution, for example, inorganic nanoparticles, carbon nanostructures or other monomers.

The multi-functionality of nanocomposites has been extensively exploited. The synergistic effects between constituents have made these materials particularly attractive as sensing elements for biological agents, and they also enable the immobilization of bioreceptors such as enzymes, antigen–antibodies, and nucleic acids onto their surfaces for the detection of an array of biological agents through a combination of biochemical and electrochemical reactions (Table 3). Some nanocomposites containing ICPs and their bioapplications will be discussed in the following paragraphs.

## 3. Bioapplications of Conducting Polymers, Conductive Hydrogels and Their Composites

Herein, we provide a brief review of recent reports on the bioapplications of ICPs, CPHs and their composites. Currently, these materials have unlimited applicability, and we have limited our discussion to three areas in which it seems that the use of ICPs is particularly interesting (Table 2). Some advantages and limitations of ICPs are summarized based on their applications.

### 3.1. Biosensors (Enzymatic Electrochemical Sensors)

Biosensors are analytical devices in which biological sensing elements (enzymes, antibodies, nucleic acids, cells, etc.) are integrated with an electronic transducer equipped with an electronic amplifier [57] (Figure 4A). Based on the IUPAC definition, a biosensor is a self-contained integral device that is capable of providing specific quantitative or semi-quantitative analytical information using a biological element [57,209]. Briefly, biosensors are chemical sensors in which the recognition system utilizes a biochemical mechanism [209]. The biological recognition system translates information from the biochemical domain into a physical or chemical output signal [209] (Figure 4A).

Biosensors may be classified by taking into account the biological specificity-conferring mechanism or the mode of signal transduction (detection or measurement mode). The first group contains a biocatalytic recognition element, in which reactions are catalysed by macromolecules, such as enzymes, cells or tissues, or contain a biocomplexing or bioaffinity recognition element, in which operation is based on antibody-antigen interaction or receptor-antagonist-agonist interaction [209]. The main parts of a biosensor and their modes of operation are depicted in Figure 4A [210]. A biosensor has two main components: an enzyme that is immobilized on the surface and a transducer. When ICPs are used in biosensors, the ICPs act as electronic transducers. Briefly, the biological part is either integrated or closely associated with the physical transducer [210]. It behaves as a recognition element, which may detect a specific biological analyte. Once the interaction takes place, the biochemical signal generated is converted, and its intensity is directly or inversely proportional to the analyte concentration. The interactions with the analyte introduce changes in the physicochemical properties resulting from the structural reorganization of polymer [212].

The first biosensing device was described almost 60 years ago by Clark and Lyons [213], who integrated enzyme and glucose oxidase (GOD) into an electrode. Since then, much progress has been made in the development of biosensors for use in diagnostic detection and monitoring of biological metabolites. Authors have focused on highly sensitive, selective and precise analytical tools for real-time estimation of biological important analytes, such as glucose, pathogens, toxins, antibodies, hormones, drugs, viruses, etc. The abundance of published reports that describe electrochemical biosensors of various designs, using both undoped ICPs, CPHs and composites containing ICPs, has increased. Just a few representative examples are described below and presented in Table 3.

The use of ICPs in sensor technologies involves employing ICPs as electrode surface modifiers to improve their sensitivity and selectivity and to suppress interference. In biosensors of different kinds, ICPs are used as active, sensing or catalytic layers and as matrices entrapping biologically active compounds [214,215]. The key process in designing a biosensor system is immobilization of the transducer, which may be achieved by adsorption, covalent bonding, entrapping or cross-linking. ICPs are frequently used in biosensors due to the opportunities they present for tuning their bio-catalytic properties and rapid electron transfer (ET) (Figure 4B,C) [211]. For the successful design of an efficient ET pathway in a biosensor, one has to consider several aspects, including (i) choice of a suitable redox mediator, (ii) design of an appropriate ET mechanism, (iii) prerequisites for fast ET, and (iv) sensor architecture [216]. These conditions are met in the case of ICPs with appropriate preparation (chemical or electrochemical) and device design.

The aim of a biosensor is to detect biologically active species as target molecules (Table 3). One of the most frequently studied molecules is glucose, which is essential for the management of diabetes. ICP-based glucose sensors provide promising solutions as they may easily accommodate GOD and give opportunities to use electrochemical techniques for target molecule detection. In a typical glucose sensor, GOD catalyses the oxidation of glucose in the presence of O_2_ to produce b-gluconic acid and hydrogen peroxide (H_2_O_2_) followed by the following reaction [217]:(1)$$\beta -\mathrm{glucose}+O_{2}+H_{2}O\;\to \;\beta -\mathrm{gluconic}\;\mathrm{acid}\;+H_{2}O_{2}$$

Presently, horseradish peroxidase (HRP) electrochemical detection of H_2_O_2_ is also possible [218]. H_2_O_2_ combines with the HRP enzyme to form an HRP compound, followed by the reduction of this compound to the original HRP [218] as follows:(2)$$\mathrm{HRP}\;\left(\mathrm{Red}\right)+2H_{2}O_{2}\to \;\mathrm{HRP}\;\mathrm{compound}\;\left(\mathrm{Ox}\right)+2H_{2}O$$
(3)$$\mathrm{HRP}\;\mathrm{compound}\;\left(\mathrm{Ox}\right)\;+2e^{-}\;+2H^{+}\;\to \;\mathrm{HRP}\;\left(\mathrm{Red}\right)+H_{2}O$$

Another important biological molecule is a quaternary protein with important medical significance, l-lactate, which is the anion that results from the dissociation of lactic acid. It is an intracellular metabolite of glucose [211]. l-lactate is the end product of anaerobic glycolysis, the final step of which is conversion of pyruvate to lactate by the enzyme lactate oxidase (LOD) or lactate dehydrogenase (LDH). The principal of an electrochemical biosensor based on LOD (Equations (4) and (5)) or LDH (Equations (6) and (7)) is described by the following reactions:(4)$$L-\mathrm{lactate}+O_{2}\overset{\mathrm{LOD}}{\to }\;\mathrm{pyruvate}+H_{2}O_{2}$$
(5)$$H_{2}O_{2}\to \;O_{2}\;+2H^{+}\;+2e^{-}$$
(6)$$L-\mathrm{lactate}+{\mathrm{NAD}}^{+}\overset{\mathrm{LDH}}{\to }\;\mathrm{pyruvate}\;+\;\mathrm{NADH}+H^{+}$$
(7)$$\mathrm{NADH}\;\overset{E}{\to }\;{\mathrm{NAD}}^{+}+H^{+}+e^{-}$$

Electrochemical cholesterol biosensors are mostly based on amperometric detection [219,220]. Cholesterol oxidase (ChOx) catalyses the biochemical degradation of cholesterol in the presence of O_2_, yielding cholest-4-en-3-one and H_2_O_2_ [220] as follows: (8)$$\mathrm{cholesterol}+O_{2}+\mathrm{ChOx}\;\to \;\mathrm{cholest}-4-\mathrm{en}-3-\mathrm{one}+H_{2}O_{2}$$

After catalytic reaction of the enzyme, H_2_O_2_ is electrochemically determined at a specific potential [220].

Enzymes are immobilized in an electroactive layer using different strategies, including physical adsorption, membrane confinement, covalent binding, cross-link formation, electrical polymerization and finally monolayer formation by self-assembly [221,222,223,224]. Notably, the physical adsorption method relies on non-specific physical interactions between the enzyme protein and the surface of the matrix, and the fundamental interacting forces are weak interactions such as van der Waals, hydrogen bonding and salt linkages [211]. This immobilization technique is not a reproducible and reliable method because of the problems associated with leaching during long-term storage of biosensors.

In the last few years, increasing analytical performance of biosensors was achieved by introducing highly porous structures, CPHs, into these devices. The micro- and mesoporous structures of CPHs offer a greater effective surface area than bulk materials, which is essential for increasing the quantity of immobilized enzyme and enhancing biosensor sensitivity. The potential advantages include the following [204]:(i)the combination of solvation and diffusion enables CPHs to be permeable to water-soluble biochemicals and chemicals,(ii)the good biocompatibility of CPHs promotes the immobilization of biomolecules,(iii)the conductivity of CPHs facilitates rapid ET,(iv)the 3D structures of CPHs favour efficient charge collection.

Biosensors formed from CPHs with entrapped enzymes, prepared by the electropolymerization of ICPs from aqueous solution, have been commonly used to prepared sensing electrodes. In these types of sensors, PANI or PPy, prepared as a hydrogel and used as a sensing layer, showed excellent selectivity even in the presence of interferents (e.g., ascorbic acid, uric acid, acetaminophen). Some representative examples are presented in Table 3.

It has been found that ICPs are rapidly degraded in the presence of H_2_O_2_. The doping of ICPs with other molecules or polymers increases their mechanical stability in terms of the electrochemical stability of biosensors and increases their analytical performance. Significant increasing biosensor stability was observed by Krzyczmonik and co-workers [225]. Electrodes modified by PEDOT doped by polyacrylic acid (PAA) and poly(4-lithium styrenesulfonic acid) (PSSLi) or poly(4-styrenesulfonic acid) (PSSH) give the best results in terms of glucose oxidation current and stability, with a long shelf life (up to 20 days).

An amperometric cholesterol biosensor was constructed based on ChOx immobilized on a conducting 4-(4H-dithienol[3,2-b:2′,3′-d]pyrrole-4)aniline (DTPPy(aryl)PPyA) polymer [220]. Glassy carbon electrodes (GCE) were modified with DTPPy(aryl)PPyA and ChOx following the procedure presented in Figure 5. Analytical performance, such as linear range (2.0–23.7 μM), detection limit (0.27 μM), limit of quantification (0.82 μM) and the Michaelis-Menten constant (*K_m_*) (17.81 μM), of the biosensor electrodes was determined. The biosensor showed good reusability and long-term stability, indicating good orientation of the enzyme on the electrode surface and the biocompatible character of the polymer.

Improvement of analytical performance was also observed by Kulesza and co-workers for an amperometric biosensor utilizing the electrocatalytic reduction of H_2_O_2_ at a GCE modified with a composite film of Prussian Blue and PPy derivative (4(pyrrole-1-yl) benzoic acid), overcoated with covalently bound GOD [230]. The addition of a functionalized organic component to Prussian Blue permits permanent attachment of an enzyme, exhibits an overall stabilizing effect, and supports fast ET (acting as a mediator, see Figure 4B) within the PPy_der_/Prussian Blue film. The proposed biosensor permitted reproducible and reliable determination of glucose in real (i.e., pharmaceutical and food) samples. The LOD and sensitivity were 1 × 10^−5^ mol L^−1^ and 2.5 μA cm^−2^ mM^−1^, respectively.

A similar concept was applied for the construction of an amperometric biosensor containing PEDOT/Prussian Blue as an active layer and HRP [231]. High electrocatalytic activity of the enzyme electrode was increased by immobilization of N-coordinated Fe(III,II) complexes capable of fast ET in the PEDOT/Prussian Blue layer (additional mediator, Figure 4B). The modified electrode responded very rapidly and produced a steady-state signal within less than 5 s. The LOD was 3 × 10^−5^ mol L^−1^.

Efforts to improve sensitivity have involved the incorporation of organic and inorganic nanoparticles into the ICP matrix. For instance, Chan Hee Park, Cheol Sang Kim and their co-workers prepared a bio-nanohybrid material based on PPy_hydrogel_ with incorporated functionalized multiwall carbon nanotubes (*f*MWCNTs) in Nafion (Nf), followed by one-step in situ electrochemical polymerization (Figure 6A) [238]. The enzymatic PPy_hydrogel_/*f*MWCNT/Nf electrode, which was prepared following the steps presented in Figure 6A, showed good selectivity, stability and excellent electrocatalytic performance to detect glucose with a high sensitivity (54.2 μA mM^−1^ cm^−2^) in a linear range of up to 4.1 mM as well as a low detection limit of 5 μM (S/N = 3) [238].

Recently, the preparation of a multidimensional conductive nanofilm composed of vertically oriented carboxylic polypyrrole nanowires (PPy_CNW_) and a graphene (G) layer was reported by Jun Seop Lee and Jyongsik Jang and their co-workers (Figure 6B–G) [239]. The conductive composite was formed using electropolymerization of Py on a G surface, followed by acid treatment. Amine-functionalized serum hepatitis B antigen (HBsAg) was immobilized on the PPy_CNW_/G surface through covalent bond formation. The sensor was highly sensitive to Hepatitis B virus (HBV) with a wide linear HBsAg concentration detection range from 10 aM to 0.1 M. The LOD was as low as 10 aM for interfering biomolecules (bovine serum albumin, immunoglobulin G, ascorbic or uric acids) with various deformations.

An interesting bioapplication of multidimensional hybrid nanocomposites was presented by Kwang-Pill Lee and co-workers [236]. They prepared poly(*N*-[3-(trimethoxy silyl)propyl]aniline (PTMSPA) and HRP on gold nanorods (Au_nanorod_) to yield PTMSPA/Au_nanorod_, which was used as a H_2_O_2_ sensing electrode. Direct electron transfer was achieved at this electrode with an ET rate constant (*k*_s_) of 3.2 ± 0.1 s^−1^. The amperometric response of the PTMSPA/Au_nanorod_/HRP modified electrode showed a quick response (<5 s) for H_2_O_2_ reduction with a wide linear range from 1 × 10^−5^ to 1 × 10^−3^ M and an LOD of 0.06 μM (S/N = 3). The electrode also exhibited high selectivity and sensitivity (0.021 μA μM^−1^) towards H_2_O_2_.

Inorganic nanoparticles incorporated into CP layers also improve the analytical parameters of biosensors [240,241,242]. A sensitive electrochemical biosensor for the detection of l-glutamate, based on immobilization of glutamate oxidase (GluO*x*) onto a PPy/zinc oxide nanorod (ZnO_NR_) composite, was described [240]. The biosensor showed an optimum response at pH 8.5 (0.1 M Tris-HCl buffer) and 30 °C when operated at 20 mV s^−1^. The biosensor exhibited excellent sensitivity (LOD of 0.18 nM), a fast response time (less than 5 s) and a wide linear range (0.02–500 μM). The enzyme electrode lost 30% of its initial activity after 100 uses over a period of 90 days when stored at 4 °C. Another application of an inorganic component in biosensing was presented by Yibing Xie and Ye Zhao [241]. GOD was immobilized onto PPy/TiO_2_ nanotubes (TiO_2NT_) via a glutaraldehyde cross-linker (Figure 6H–K). The hydrophilic PPy/TiO_2NT_ hybrid provided highly accessible nanochannels for GOD encapsulation, presenting good enzymatic affinity. The enzyme electrode nicely conducted bioelectrocatalytic oxidation of glucose, exhibiting good biosensing performance with a high sensitivity (187.28 μA mM^−1^ cm^−2^), low detection limit (1.5 M) and wide linear detection range. An organic-inorganic composite containing PPy (with different contents) and Co_3_O_4_ provided the opportunity to incorporate haemoglobin (Hb) and GOD and influence the direct ET of Hb/GOD [242] (Figure 6H–K). Notably, when the weight percentage of the Py monomer was 20%, the heterogeneous ET *k*_s_ values for Hb and GOD were estimated to be 1.71 and 1.67 s^−1^, respectively. The composite modified electrode was used as a biosensor and exhibited a long linear range and low LOD (0.71 μM) to H_2_O_2_. The sensing design based on the PPy/Co_3_O_4_ hybrid material was demonstrated to be effective and promising for the development of protein and enzyme biosensors.

The unique property that ties the biosensing of ICPs is their conductivity. ICPs are organic in nature, making them biocompatible. The easy preparation and modification of ICPs have made them a popular choice for biosensors. Despite the vast amount of research conducted in this area, the field is still growing, and many questions remain to be answered.

### 3.2. Main Assumptions of Tissue Engineering and ICP Applications in This Area

The term “tissue engineering” was used officially for the first time in 1988 at a National Science Foundation workshop. It was created to represent a new scientific area for tissue regeneration from cells using biomaterials, scaffolds and growth factors [56]. TE, also known as regenerative medicine, is a field of technical sciences that uses medical knowledge and material engineering methods. TE provides new medical therapies as an alternative to conventional transplantation methods. TE regulates cell behaviour and tissue progression through the development and design of synthetic extracellular matrix analogues of novel biomaterials to support 3D cell culture and tissue regeneration [160]. The aim of TE is to replace, restore, improve or maintain the function of tissues and organs using implants containing the patient’s own cells embedded in a special material that acts as a scaffold for cells [24,33].

Cells grown in vitro are placed on such scaffolds, and the whole construct is implanted in the patient’s body in place of the defect [243]. Scaffolds for TE are prepared from appropriate biocompatible materials, which in the in vitro stage form a physical basis for cells and act as an artificial intercellular substance stimulating cells to multiply, differentiate and reproduce the desired tissue [244]. In a second in vivo stage, new tissue is formed, and the materials used as the scaffold are gradually degraded and resorbed by the body [33]. Due to this process, there is no need to carry out surgery to remove unnecessary scaffolding, and the reconstructed new tissue is identical to healthy tissue, both in terms of function and structure. Therefore, intensive research is being carried out on preparation methods and scaffold properties that can be used in TE [245].

When designing a scaffold for use in TE, the following key issues are important [24,246,247,248]:(i)biocompatibility to prevent an inflammatory reaction that may cause rejection by the body;(ii)biodegradability of scaffolds that are not intended as a permanent insertion;(iii)mechanical properties should be adequate to the tissue into which it is to be implanted;(iv)scaffold architecture is a frame that facilitates the creation of tissue;(v)the type of biomaterial from which the scaffold is fabricated and the manufacturing technology should be clinically and commercially viable.

These issues are met in the case of ICPs [154]. CPs are attractive biomaterials for TE applications as they can deliver electrical signals to cells for the regeneration of injured tissues. Their greatest advantage is their vast versatility [56]. The key to this is the dopant, which defines CP applicability [56].

Due to their ability to electronically control a range of physicochemical properties, CPs and their composites provide compatible substrates that promote cell growth, adhesion, and proliferation at the polymer—tissue interface [32,154]. Specific cell responses depend on CP surface characteristics such as roughness, topography, surface free energy, charge, chemistry, and other properties such as electrical conductivity or mechanical actuation [32,249].

In recent years, it also has been indicated that nanostructured biomaterials significantly affect the biological functions of cells [250,251,252,253,254]. Within this group of materials at the nanometre scale (nanofibres, nanotubes, nanoparticles, and nanofilaments) [254,255], CPs may be applied, and their biological activity towards cells may be controlled by electrical stimulation [32,160,256,257,258,259,260]. These properties depend mainly on the employed synthesis conditions. Micro- and macrostructured CPs may be synthesized using an electrohydrodynamic method called flow-limited field-injection electrostatic spraying [253], electrospinning [254,257,258,259], plasma polymerization [260] and wet-spinning [261], etc. Manipulation of synthesis parameters also offers fine control over the physical properties of CPs, which is a useful tool for tuning the material for specific TE applications (Figure 7) [160,249].

#### Conductive Hydrogels in Tissue Engineering

The fact that hydrogels are biomaterials designed specifically for human use is what makes hydrogels good potential candidates for tissue scaffolds [173,262] used in tissue regeneration. A special role is played by CPHs that have electrical conductivity ability, which provides increased cell growth, adhesion and proliferation during the regeneration of muscle, cardiovascular nerves, and bone tissue cells [56,186,249,263,264]. A hydrogel may be synthesized in few different forms—as a copolymer, a graft copolymer or a composite—and each variety will have a different structure [159].

An example of a single component CP is PTAA (poly(3-thiopheneacetic acid)) hydrogel, characterized for potential use as a scaffold in biomedical applications [234]. The PTAA polymer has a porous internal structure, good mechanical properties comparable to those of muscle tissue, appropriate adhesion and increased myoblast cell proliferation. These characteristics offer the possibility of using PTAA in TE as scaffolds for nerve and muscle regeneration.

CPHs mainly consist of two components: a CP component responsible for electrical conductivity and a hydrogel unit that ensures environmental hydration. The first such material was created based on PPy electropolymerized straight onto a preformed polyacrylamide (PAAm) hydrogel [265]. From that moment, there was a rapid development of CPH applications in TE [173,199,266,267,268,269]. The most commonly used ICPs, among all CPs, are PPy, PT, PEDOT and PANI [270]. The most frequently used in a wide range of biomedical fields are PANI and PPy. The use of these ICPs in artificial muscles, TE, and drug delivery systems derives from the fact that they are biocompatible, chemically stable, and their synthetic process and doping are simple [271,272,273,274]. Certain tissues, such as muscle, may require different material properties, as these tissues need flexibility as a fundamental part of their mode of action (Figure 7). To address this, modification of a flexible polymer was conducted to make it more suitable for the culture of muscle cells [275].

CPHs are used in TE to imitate the specific properties of cardiac tissue, which has significantly limited regenerative capacity [276,277]. An example of the use of CPHs in cardiac tissue engineering is a homogeneous hydrogel containing PTAA and methacrylate aminated gelatin. The conductivity of this hydrogel (10^−1^ mS·cm^−1^) is similar to the conductivity of myocardial tissue. Biological evaluation of this hydrogel showed its positive effects on cardiac differentiation efficiency [278].

Savyar and co-workers prepared conducting biocompatible PPy/graphene/chitosan composite hydrogels (Table 4) [268]. The addition of 3% graphene caused an increase of over 200% in terms of the mechanical properties (tensile strength) and conductivity of these hydrogels. Fibroblast cells showed good adhesion, proliferation and viability on the surfaces of these composites, indicating that they are excellent candidates for biodegradable materials in TE cell scaffolds with very low risk of graphene accumulation during degradation. In turn, the hydrogel affects the increase in cell viability and proliferation of human embryonic stem cells derived from fibroblasts and cardiac muscle cells [279,280].

An example of a CPH with high potential for use as a conductive substrate for the growth of cells responsive to electricity in TE is a conductive polymer made by Y. S. Kim and co-workers [292], consisting of PEDOT:PSS poly(4-styrenesulfonic acid) in a PEG (poly(ethylene glycol)) hydrogel matrix (Table 4). This CPH can be used as an electrically conductive TE scaffold for muscle and nerve tissue. The surface of the above-mentioned CPH can be easily modified to aid cell adhesion and proliferation, and in vitro studies conducted using electrically responsive H9C2 myocytes showed that this hydrogel is not cytotoxic. Sasaki and co-workers reported the fabrication of the hydrogel-based devices that remain highly electrically conductive properties using a combination of chemical polymerization and electropolymerization of PEDOT and polyurethane (PU) [185]. The PEDOT/PU–hydrogel hybrids exhibited a high electrical conductivity of up to 120 S cm^−1^ at 100% elongation. The CPH was biocompatible with both muscle and neural cells. The biosynthetic co-hydrogel containing PVA–heparin and PEDOT with incorporated sericin and gelatin were prepared via methacrylate crosslinking and electrochemical polymerization [184]. Electrical properties of the bioactive CPH were assessed, and it was shown that the bioactivity of heparin was retained after electropolymerization of ICP through the hydrogel. Similarly, incorporation of sericin and gelatin in the CPH promoted neural cell adhesion and proliferation. The adhesion, proliferation, and differentiation of neural and muscle cells cultured on these hybrids are demonstrated, as well as the fabrication of 3D hybrids, advancing the field of tissue engineering with integrated electronics. The conducting PPy composite hydrogels possessed very good swelling/deswelling potential as well as good electrical conductivity, indicating they can be used in TE [293].

One very interesting combination of polymers are the composites containing CPs and stimulus-responsive polymers [287]. The latter are a special class of polymeric materials that can respond to even very slight changes in temperature, pH, light, and ionic strength, permitting their wide use in TE, drug delivery systems and sensors [287]. Temperature-sensitive materials have attracted significant attention owing to their ability to display intelligent responses to temperature changes. Sharma used poly(N-isopropylacrylamide) (PNIPAM), which may act as a controllable temperature-responsive bio-switch, and PANI for controlled cell adhesion. When fibroblast cells were seeded on the composite surface, the PANI/PNIPAM nanofibres exhibited the highest cell growth and a %live of approximately 98%, indicating very good biocompatibility.

Ongoing studies seek to generate new CPHs that would be biocompatible with cells of the human body and would aid in the treatment and rehabilitation of damaged tissues sensitive to electrostimulation [293].

### 3.3. Drug Delivery Systems Containing ICPs

The family of electroactive biomaterials is considered a new generation of smart materials that allow the direct delivery of electrical signals by converting their chemical, electrical and physical properties (Figure 8A) [202]. These biomaterials include CPs, piezoelectrics, photovoltaic materials, and electrets. Their unusual properties result in unlimited applications (e.g., medicine, pharmacy and agriculture). CPs are perfect materials for the controlled delivery of chemical compounds [170,239,294], mainly in CPH form [295].

Since the 1980s, when Zinger and Miller demonstrated that glutamate and ferrocyanide can be released from PPy films through the application of an electric potential [296], CPs have been investigated as potential candidates for drug delivery systems. Briefly, drug delivery systems based on CPs exploit the polymers’ ability to electrically switch between an oxidized and a reduced state.

The desired molecules incorporated into the CPH matrix by using a doping method can be released in a controlled manner after applying a reducing (negative) or oxidative (positive) electrical potential [56,239,281,294]. This creates a flexible, lightweight and partially biodegradable device that does not require an external power source to operate. The diffusion of bound molecules is facilitated by the porous structure of CPHs and the presence of delocalized charge carriers in their structure [239]. This theory has been confirmed by studies conducted for medicinal substances such as dopamine [297], naproxen [298], heparin [281,299] and dexamethasone [294]. The discharge of particles usually takes place quite rapidly, within just a few minutes, which, depending on the CPH used, can be both a disadvantage and an advantage of CPs [299,300,301].

Loading of the drug compound can be performed in a number of ways depending on the type of drug, determined mainly by its size (small or big) and charge (positively and negatively charged or neutral drug compounds) (Figure 8B) [202]. Small anionic compounds can be loaded through one-step immobilization (Figure 8B(1)) as dopants during the polymer synthesis process [302,303]. When the drug molecules interfere with the polymerization process, the most complicated loading process is required. The three-step method separates the synthesis and drug loading processes (Figure 8B(2)) [202,302,303,304,305], which comprise the following: (i)the synthesis of polymer using an ideal anionic “primary” dopant;(ii)application of a reducing potential to flush out the primary dopant;(iii)incorporation of the desired medicinal compound into the polymer by reversing the potential [202].

Using this method, large anionic compounds may be incorporated in the polymer matrix. Cationic drugs require a modified version of the three-step method, which is presented in Figure 8B(3) [202]. A large primary anionic dopant is immobilized inside the polymer matrix during synthesis. Next, a reducing potential is applied to the polymer, which results in the positively charged drug entering the material to maintain electroneutrality.

The limitation of the use of CPs for the transport and release of drugs is the diffusion of bound molecules outside the polymer and replacement of these particles with others found in the polymer environment, as well as a small amount of the drug bound in the polymer. Molecular weight and charge also determine which molecules can be bound and released. In this respect, modification of CP chains is frequently required. To remove such limitations, various modifications are made, e.g., biotin-streptavidin conjugation with PPy [300,306]. Biotin is a dopant that provides more homogeneous kinetics of release. As a result of the application of an electrical stimulation, the bioactive molecule, covalently attached to biotin, is released. This is due to reduction of the PPy backbone through the use of electrical stimulation and release of biotin [56,300,301].

Saha and co-workers described the preparation and use of a CPH based on PAAM and PPy [289]. This hydrogel, obtained by electrochemical polymerization of PPy into PAAM and assuming the form of an implantable, cylindrical drug-delivery device, was proposed for the treatment of schizophrenia and bipolar disorder. The controlled release of model compounds, specifically safranine [307] and methylrosaniline chloride [180], was tested with PPy/PAAM hydrogels developed by electropolymerization and proved their suitability for use in electrochemically controlled drug release processes. It is extremely important that this type of hydrogel is ‘active’ under neutral pH conditions, which significantly increases its attractiveness for in vivo applications. The shortened lifespan of CPHs can be seen in PPy/PPS composites whose conductivity drops by 95% after 16 hours of 0.4 V voltage application [56]. As a result of repetitive stimulation cycles, the CPH undergoes irreversible polymer oxidation reactions and a parallel process of dedoping and reduction of conductivity, shortening the lifespan of the CPH. PPy and PEDOT very often are combined and synthesized in hydrogels in pristine form or by cross-linking with other polymers, which can be individually used in the drug delivery process. These polymers include, for example, alginate [201], xanthan [284] and poly(lactic-*co*-glycolic acid)-*co*-poly(ethylene glycol) (PLGA-PEG) [308]. Such CPHs are characterized by electrical conductivity, swelling ability and biocompatibility [309].

Poly(*p*-phenylenevinylene) (PPV) was used to create a hydrogel with PAAM. This combination of components resulted in the delayed release of salicylic acid from 3 to more than 15 h after application of the appropriate anode potential. The release profile of salicylic acid was optimized by using a cathodic potential, different electric field strengths as well as cross-linking densities, hydrogel pores and drug molecule sizes [133,202]. A PAAM and PANI hydrogel was also used in a drug delivery system [288].

Notably, for many applications, simple CP films alone do not provide sufficient drug storage capacity. The use of organized porous structures (micro- and nanostructures) of CPs can provide a greater volume and surface area for drug binding (Figure 8C) [202]. PPy nanowires were formed, and the micro- and nanogaps between the wires served as reservoirs for the binding of ATP and dexamethasone (Figure 8C(1)) [202,310]. Chemically synthesized PANI nanofibres loaded with amoxicillin were encapsulated into a PAAM hydrogel and evaluated to establish release and toxicity profiles. Research confirmed the attractiveness of this hydrogel for electrically controlled drug delivery applications, including implantable devices and transdermal drug delivery systems. Micro- and nanotubes of PEDOT were used to load bacterial cellulose [311], PLLA or PLGA [312] (Figure 8C(2,3)).

Nanocomposites with incorporated nanoparticles (NPs) offer an additional solution to improve drug-loading capacity. ICPs with NPs show an increased specific surface area, which results in increased encapsulation/loading efficiency of the drug [202,313,314,315,316]. Graphene oxide (GO) was combined with PPy to generate a composite material with twice greater dexamethasone binding capacity than PPy alone, a linear release profile up to 400 stimulations, and no passive drug diffusion [313]. GO was also combined with PEDOT and as a composite to deliver dexamethasone in a smart coating for orthopaedic implants [314]. An encapsulation efficiency of 95% was achieved when loading ketoprofen inside PPy-iron oxide nanoparticles (Figure 8C(4)) [202]. The PPy nanoparticles were also immobilized in a calcium alginate hydrogel for the sustained pH-dependent release of the anti-inflammatory drug piroxicam [317,318]. Rapamycin was bound in a liposome wall formed around PPy nanoparticles [319]. The liposome was coated with Herceptin^®^, which binds specifically to a receptor expressed by breast cancer cells. Exposing the cells to an 808-nm laser heated up the particles, releasing rapamycin and triggering apoptosis [319]. The combination of the conductive properties of CPs and hydrogel capabilities allows them to be used for various biomedical purposes [320], including as drug delivery systems applied directly to a target area.

## 4. Conclusions and Future Prospects

To date, an enormous number of approaches have been developed to formed ICPs, CPHs and their composites due to their further applications in biomedicine. The most common ICPs for these purposes are PANI, PPy, PEDOT, PT, PA, PPP and their derivatives. In this review, we highlighted some aspects of the structures of ICPs, their composition, roughness and other physicochemical properties resulting in their application in biomedical areas. Some limitations of ICPs, such as a low processability, their hydrophobic character, poor mechanical properties and poor biocompatibility, have forced researchers to explore new methods, that modified physicochemical properties of ICPs. However, our review showed that ICPs have been used for different biomedical purposes including drug delivery systems, tissue engineering, biosensors, further important progress is required before using them in commercialized health systems.

Recently, there has been increasing worldwide demand for the development of molecular-sized materials for a variety of applications, in particular the preparation and integration of multifunctional molecules into ordered supramolecular architectures. However, although the techniques for functionalizing organic molecules are well known, there are still many problems with controlling the physical and chemical properties of “higher” architectures. Among these materials, we can find highly hydrophilic porous materials such as hydrogels. Different organizations of the gel network result in different physicochemical properties and functionalities. In this respect, controlling the structure of the framework by controlling the organization of the pores and their sizes is required for further applications of these materials in biomedicine.

Hydrogels can also act as a ‘host network’ for the immobilization of other materials inside, such as ICPs and/or biomolecules, resulting in CPHs or their composites. Although some physicochemical properties were tuned in these hybrid porous hydrogels, the main disadvantages of these materials have been their non-uniformity and random composition due to the low compatibility of both components. Incorporation of biological molecules or ICPs into hydrogels, changes their mechanical and electrical properties, their biocompatibility and capacity. The key challenge in forming CPHs is to preserve the overall electroactivity with fine-tuning the desired mechanical softness, elasticity and biocompatibility. The application of hydrogels is closely related to their three-dimensional porous structures, and this is determined by the conditions of their formation. Well-designed three-dimensional hydrogels may serve as the basis for next-generation drug delivery systems, tissue engineering, biosensors, all of which exhibit significant performance advantages over current state-of-the-art technologies. Even so, CPHs still have many limitations. In view of the above, it seems that the search for new hydrophilic porous materials that can be formed by simple technologies requires continuous basic research and remains a critical challenge.

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
