# Peer review of "Conducting Polymers, Hydrogels and Their Composites: Preparation, Properties and Bioapplications"

_polymers, 2019, doi:10.3390/polym11020350_

Round 1

Reviewer 1 Report

- Conducting polymers are mostly hydrophobic. So, give more information on what strategies can be used for using conducting polymers in the hydrogel matrix (not just physical mixing or physical incorporation). More specifically, address more about how the effective surface area of conducting polymers can be enhanced or how the interfacial tension between aqueous media and conducting polymers. 

- As the authors know, there are already many review articles particularly regarding conducting polymer-based biosensors. It's difficult for me to find meaningful difference in this manuscript. To be published, it's very important to add more distinctive contents.

- Additionally, there are many review articles regarding hydrogels and tissue engineering (for example, Polymers, 2017, 9, 364; Gels, 2018, 4, 75; Gels, 2017, 3, 25, and so forth).

- Lastly, there are many relevant articles regarding conductive polymers, not cited in the manuscript: for example, line 43/line 46/line 210 electrical properties/charge transport (Polymers, 2017, 9, 150); line 56 sensors (Polymers, 2017, 9, 155); line 288 nanoparticles (Polymers 2016, 8, 118).

- As a minor point, Figure 2 should be colored or re-made for clarification.  

Author Response

Reviewer #1:

Comment: Extensive editing of English language and style required.

Please find Editorial Certificate from ACS Chemworx Authoring Services for this review.

Comment: Conducting polymers are mostly hydrophobic. So, give more information on what strategies can be used for using conducting polymers in the hydrogel matrix (not just physical mixing or physical incorporation). More specifically, address more about how the effective surface area of conducting polymers can be enhanced or how the interfacial tension between aqueous media and conducting polymers. 

In response to this reviewer request, we added some explanations on Page 7 and 8.

Comment: As the authors know, there are already many review articles particularly regarding conducting polymer-based biosensors. It's difficult for me to find meaningful difference in this manuscript. To be published, it's very important to add more distinctive contents.

We agree with the reviewer. The number of publications regarding this issue is huge. It is always a lack of satisfaction that some issues have been briefly discussed or omitted. The section regarding biosensors is already the most expanded in our review. The review was intended only to highlight the key issues of some areas of biomedicine, therein biosensors. A few literature references suggested by the reviewers were added. Because of the length of this manuscript, the authors restricted to additional expansion and omitted additional explanation in this paragraph.

Comment: Additionally, there are many review articles regarding hydrogels and tissue engineering (for example, Polymers, 2017, 9, 364; Gels, 2018, 4, 75; Gels, 2017, 3, 25, and so forth).

Requested references were added.

Comment: Lastly, there are many relevant articles regarding conductive polymers, not cited in the manuscript: for example, line 43/line 46/line 210 electrical properties/charge transport (Polymers, 2017, 9, 150); line 56 sensors (Polymers, 2017, 9, 155); line 288 nanoparticles (Polymers 2016, 8, 118).

As above. Requested references were added.

Comment: As a minor point, Figure 2 should be colored or re-made for clarification.  

In response to this reviewer request, we did corrections of Figure 2. Figure 2 was re-made and some additional our results (not published) were added as Panel B. The corresponding comments have also been added in the text on Page 7.

Reviewer 2 Report

This review article well collect research facts on important subjects of conductive polymers. This informative review is valuable for publication in Polymers. Therefore, I basically recommend publication of this review in Polymers. However, some parts of this review have to be improved. Please see below for necessary revisions.

1) The most problematic point of this review is lack of clear section of Conclusion and Perspective. This manuscript concludes this manuscript ambiguously with a small paragraph. Because this review itself has rich information, such small conclusion does not well fit with its main contents. It is highly recommended to make clear section of Conclusion (or Conclusion and Perspectives) at the last of manuscript. In addition, detailed and hot descriptions for future perspectives on conductive polymers had better be added in this conclusive sections.

2) Visual items in this review article are only copied from the previous publications. Therefore, own opinions by the authors were not well emphasized in this review. I recommend to add two own figures, (i) backgrounds and motivation at Introduction and (ii) summary and perspective at Conclusive sections.

3) Reference selection is rich but initial part needs citations of rather recent papers on conductive polymers (for example, Bulletin of the Chemical Society of Japan 2017, 90(12), 1388-1400; Joule 2017, 1(4), 739-768; Nano Energy 2018, 44, 265-271; Bulletin of the Chemical Society of Japan 2017, 90(7), 847-853; Chemical Reviews 2018, 118(14), 6766-6843).

4) Information on conductive polymers are not provided in systematic ways. For example, all the chemical structures given in Table 1 are better to be given.

Author Response

Referee Letter: This review article well collect research facts on important subjects of conductive polymers. This informative review is valuable for publication in Polymers. Therefore, I basically recommend publication of this review in Polymers. However, some parts of this review have to be improved. Please see below for necessary revisions. The most problematic point of this review is lack of clear section of Conclusion and Perspective. This manuscript concludes this manuscript ambiguously with a small paragraph. Because this review itself has rich information, such small conclusion does not well fit with its main contents. It is highly recommended to make clear section of Conclusion (or Conclusion and Perspectives) at the last of manuscript. In addition, detailed and hot descriptions for future perspectives on conductive polymers had better be added in this conclusive sections. Visual items in this review article are only copied from the previous publications. Therefore, own opinions by the authors were not well emphasized in this review. I recommend to add two own figures, (i) backgrounds and motivation at Introduction and (ii) summary and perspective at Conclusive sections.

Comment: In response to this reviewer request, we added Conclusion and Perspectives, where our opinion are highlighted. We corrected Figure 2 (Page 2), where our results (not published) were provided.

Comment: Reference selection is rich but initial part needs citations of rather recent papers on conductive polymers (for example, Bulletin of the Chemical Society of Japan 2017, 90(12), 1388-1400; Joule 2017, 1(4), 739-768; Nano Energy 2018, 44, 265-271; Bulletin of the Chemical Society of Japan 2017, 90(7), 847-853; Chemical Reviews 2018, 118(14), 6766-6843).

Requested references were added.

Comment: Information on conductive polymers are not provided in systematic ways. For example, all the chemical structures given in Table 1 are better to be given.

In response to this reviewer request, the chemical structures of CPs are included in Table 1.

Round 2

Reviewer 1 Report

The authors have made effort to enhance the quality of the manuscript.